# Targeted Delivery of Epidermal Growth Factor to the Human Placenta to Treat Fetal Growth Restriction

**DOI:** 10.3390/pharmaceutics13111778

**Published:** 2021-10-25

**Authors:** Lewis J. Renshall, Frances Beards, Angelos Evangelinos, Susan L. Greenwood, Paul Brownbill, Adam Stevens, Colin P. Sibley, John D. Aplin, Edward D. Johnstone, Tambet Teesalu, Lynda K. Harris

**Affiliations:** 1Maternal and Fetal Health Research Centre, Division of Developmental Biology and Medicine, Faculty of Biology, Medicine and Health, The University of Manchester, Manchester M13 9WL, UK; lewis.renshall@manchester.ac.uk (L.J.R.); frances.beards@manchester.ac.uk (F.B.); angelos.evangelinos@manchester.ac.uk (A.E.); susan.l.greenwood@manchester.ac.uk (S.L.G.); Paul.brownbill@manchester.ac.uk (P.B.); Adam.stevens@manchester.ac.uk (A.S.); Colin.sibley@manchester.ac.uk (C.P.S.); John.aplin@manchester.ac.uk (J.D.A.); edward.johnstone@manchester.ac.uk (E.D.J.); 2Manchester Academic Health Science Centre, Central Manchester University Hospitals NHS Foundation Trust, St Mary’s Hospital, Manchester M13 9WL, UK; 3Cancer Research Center, Sanford Burnham Prebys Medical Discovery Institute, La Jolla, CA 92037, USA; tambet.teesalu@ut.ee; 4Center for Nanomedicine, University of California Santa Barbara, Santa Barbara, CA 93106, USA; 5Division of Pharmacy and Optometry, Faculty of Biology, Medicine and Health, The University of Manchester, Manchester M13 9WL, UK

**Keywords:** placenta, pregnancy, fetal growth restriction, liposomes, epidermal growth factor

## Abstract

Placental dysfunction is the underlying cause of pregnancy complications such as fetal growth restriction (FGR) and pre-eclampsia. No therapies are available to treat a poorly functioning placenta, primarily due to the risks of adverse side effects in both the mother and the fetus resulting from systemic drug delivery. The use of targeted liposomes to selectively deliver payloads to the placenta has the potential to overcome these issues. In this study, we assessed the safety and efficacy of epidermal growth factor (EGF)-loaded, peptide-decorated liposomes to improve different aspects of placental function, using tissue from healthy control pregnancies at term, and pregnancies complicated by FGR. Phage screening identified a peptide sequence, CGPSARAPC (GPS), which selectively homed to mouse placentas in vivo, and bound to the outer syncytiotrophoblast layer of human placental explants ex vivo. GPS-decorated liposomes were prepared containing PBS or EGF (50–100 ng/mL), and placental explants were cultured with liposomes for up to 48 h. Undecorated and GPS-decorated liposomes containing PBS did not affect the basal rate of amino acid transport, human chorionic gonadotropin (hCG) release or cell turnover in placental explants from healthy controls. GPS-decorated liposomes containing EGF significantly increased amino acid transporter activity in healthy control explants, but not in placental explants from women with FGR. hCG secretion and cell turnover were unaffected by EGF delivery; however, differential activation of downstream protein kinases was observed when EGF was delivered via GPS-decorated vs. undecorated liposomes. These data indicate that targeted liposomes represent a safe and useful tool for the development of new therapies for placental dysfunction, recapitulating the effects of free EGF.

## 1. Introduction

Fetal growth restriction (FGR), the failure of a fetus to reach its genetically determined growth potential, is a serious complication affecting up to 10% of pregnancies. FGR is a major risk factor for neurodevelopmental disorders and stillbirth, and the main underlying cause is placental dysfunction [1,2,3]. There are no drug treatments available, so FGR is managed by early delivery and neonatal intensive care. There has been negligible investment in obstetric therapeutics by government, charities, or pharmaceutical companies for decades [4,5], primarily due to the risks of harmful or teratogenic side effects posed by developing and testing new compounds. This is despite the large economic and global health burden of maternal and perinatal conditions, and a sizeable untapped market for these products [6].

To address this important problem and minimize the risks of systemic drug delivery in pregnancy, we have developed a way to selectively deliver drugs to the placenta. This approach prevents transfer of drugs to the fetus and dramatically reduces drug accumulation in other maternal organs. We have previously demonstrated that the peptide sequences CGKRK and iRGD selectively bind to the placenta and uterine vasculature of mice and humans, and do not alter normal fetal or placental development [7]. Liposomes decorated with the iRGD peptide containing insulin-like growth factor 2 (IGF-2) were intravenously administered to a mouse model of FGR, and IGF-2 was selectively delivered to the uteroplacental interface, leading to a significant increase in placental weight and a reduced number of growth-restricted fetuses [7]. This study demonstrated that targeting therapies directly to the placenta and uterine vasculature could provide a safe and effective way of treating placental dysfunction.

Abnormalities in placental structure or function, occurring either early in gestation, or as a result of poor uteroplacental oxygen supply and/or elevated oxidative stress mid-pregnancy, can impair the capacity of the placenta to support the developing fetus, and these abnormalities are commonly associated with adverse outcomes such as FGR and stillbirth [3,8,9,10]. Placental nutrient transfer to the fetus may be compromised in FGR, due to reduced nutrient transporter expression and/or activity [11,12]. In addition, reduced placental cell proliferation [13] and increased placental cell apoptosis [14,15] have also been observed in FGR pregnancies. To develop successful targeted interventions to treat FGR, we need to select candidate therapies which can correct one or more aspects of placental dysfunction observed in this condition. Epidermal growth factor (EGF) is a 6 kDa polypeptide that regulates cell growth and differentiation by binding to the EGF receptor (EGFR). EGF binding induces receptor dimerization, tyrosine kinase activation and initiation of a signal transduction cascade that leads to an increase in intracellular calcium, changes in gene expression, and ultimately, increased DNA synthesis and cell proliferation. EGF is an important regulator of fetal and placental growth, as embryos lacking EGFR expression exhibit defects in placental development and FGR [16]. Furthermore, decreased EGFR signaling has been reported in FGR placentas [17], and exogenous EGF has been shown to protect against cytokine- and reactive oxygen species-induced apoptosis in human placental explants cultured ex vivo [18,19,20]. Taken together, these data indicate that EGF is an important regulator of normal placental and fetal growth, and suggest that targeted delivery of EGF to the placenta in FGR pregnancies may represent a novel intervention to treat the condition.

The aim of the current study was to identify a novel placental homing peptide that bound exclusively to the outermost layer of the placenta, the syncytiotrophoblast, and did not bind to the local vasculature. We hypothesized that (i) this peptide could be used to enhance the efficacy of drug-loaded liposomes, and (ii) liposomally encapsulated EGF could improve one or more aspects of human placental function.

## 2. Methods

### 2.1. Phage Screening

All animal procedures were reviewed and approved by the Institutional Animal Care and Use Committee (IACUC) at The University of California, Santa Barbara, USA (approval code: 11-010-814; approval date: 2 December 2011). BALB/c mice (Charles River, USA) were maintained on a 12:12 h light/dark cycle at 21–23 °C and had free access to food and water. Successful mating was indicated by the presence of a copulation plug and was denoted as embryonic day 0.5 (E0.5) of pregnancy. Seven mice, with pre-pregnancy weights of 22.2–26.3 g and aged between 8 and 11 weeks were used.

CX7C peptide libraries were constructed using the T7-select phage display system (EMD Biosciences, San Diego, CA, USA). Individual phages were cloned following the manufacturer’s instructions, as previously described [21,22]. Following amplification, phages were purified from bacterial lysates by precipitation with PEG-8000 (Sigma Aldrich, St. Louis, MO, USA), caesium chloride gradient ultracentrifugation and dialysis. The amino acid sequence of the displayed peptides was ascertained by sequencing the DNA encoding the insert-containing region at the C terminus of the T7 major coat protein gp10 (Eton Bioscience, San Diego, CA, USA).

Mouse placentas harvested between E13.5 and E14.5 of pregnancy were washed in PBS and homogenized using a Medimachine system (BD Biosciences, San Jose, CA, USA). The tissue homogenates were incubated with the T7 phage library (10^10^ pfu/mg tissue) in 10 mL DMEM culture medium containing 1% (*w*/*v*) BSA (Life Technologies, Carlsbad, CA, USA) for 1 h at 4 °C on a rotating wheel. The tissue was pelleted by centrifugation (400× *g*; 5 min), washed five times in fresh culture medium (400× *g*; 5 × 5 min) and lysed in LB bacterial growth medium containing 1% Nonidet P-40 (Sigma Aldrich, St. Louis, MO, USA). Phages which remained bound to the tissue were quantified by titration and amplified, as previously described [21,22]. This ex vivo biopanning process was repeated three times, to select for phage displaying peptides which bound to the placental tissue. The resulting phage pool was injected into the tail vein of a pregnant mouse and allowed to circulate for 30 min [7]. Mice were anaesthetized with an intraperitoneal injection of Avertin (150–250 mg/kg) and were subjected to terminal cardiac perfusion with 30 mL of PBS, to remove unbound phage from the tissues. The uterus and placentas were removed by dissection and homogenized in LB bacterial growth medium containing 1% (*v*/*v*) Nonidet P-40. Bound phage were tittered, amplified and purified, as previously described. Four rounds of in vivo screening were performed in total, using pregnant mice between E13.5 and E15.5. Ninety-six individual phage clones selected from the second, third and fourth rounds of screening were sequenced to determine the amino acid sequence of their surface peptides.

### 2.2. Peptide Localization

CCGPSARAPC (GPS) peptide was purchased from Insight Biotechnology, London, UK, and was labeled with either 5(6)-carboxyfluorescein (FAM-GPS) or 5-carboxytetramethylrhodamine (TAMRA-GPS). Mice received 200 µg of peptide via tail vein injection at gestations of between E11.5 and E17.5 [7]. After 3 h, unbound peptide was removed by terminal cardiac perfusion with PBS, and maternal, placental and fetal tissues were harvested. Organs were snap-frozen in liquid nitrogen, or were fixed in paraformaldehyde (4% (*w/v*) in PBS, overnight), transferred to sucrose solution (30% (*w/v*) in PBS, 24 h), embedded in OCT (Sakura, Torrance, CA, USA) and stored at −80 °C. Snap-frozen tissues were sectioned using a cryostat, and 8 μm sections were fixed in ice-cold methanol (15 min), washed in PBS (2 × 5 min) and mounted using Vectashield mounting medium containing DAPI (4′,6-diamidino-2-phenylindole; Vector Laboratories, Burlingame, CA, USA). Peptide localization was assessed using a Zeiss AxioObserver fluorescence microscope (Zeiss, Cambridge, UK). In a separate experiment, human placental villous explant cultures were established as described below. Explants were exposed to TAMRA-GPS (0.27 µmol/L) for up to 3 h at 37 °C and were fixed and processed as above. Images were captured at the same exposure, so that comparisons of fluorescence intensity could be made between samples.

### 2.3. Liposome Formulation

The thin film method was used to prepare liposomes [7]. Targeted liposomes were composed of 1,2-distearoyl-sn-glycero-3-phosphocholine (DSPC; 65 μmol/L), cholesterol (30 μmol/L), 1,2-distearoyl-sn-glycero-3-phosphoethanolamine-*N*-[amino(polyethylene glycol)] (DSPE-PEG; 4.5 μmol/L) and DSPE-PEG-maleimide (0.5 μmol/L; Avanti Polar Lipids, Birmingham, AL, USA), and lipids were dissolved in methanol:chloroform (9:1 ratio). Non-targeted liposomes were composed of DPSC (65 μmol/L), cholesterol (30 μmol/L) and DSPE-PEG (5 μmol/L), which were dissolved in methanol:chloroform (9:1 ratio). For both targeted and non-targeted liposomes, the solvent was removed by rotary evaporation to produce a thin lipid film, which was rehydrated with human recombinant EGF in sterile PBS (2 mL, 200 or 100 ng/mL, Sigma Aldrich, Gillingham, UK). The resulting suspension was heated to 55 °C for 4 h and vortexed hourly to produce large multilamellar vesicles. The suspension was extruded 15 times using a 1 mL Mini-Extruder (Avanti Polar Lipids, Birmingham, AL, USA) through a 0.2 μm, 19 mm polycarbonate membrane, surrounded by two 10 mm filter supports in order to produce a ~200 nm unilamellar liposome suspension. To create the targeted formulation, the fluorescent GPS peptide (0.27 μmol/L), which bore a cysteine residue on its N-terminus, was added to 1 mL of extruded liposome suspension and incubated overnight at room temperature. The peptide covalently coupled to maleimide groups on the liposomal surface via a Michael-type addition reaction. Unreacted peptide was removed from the formulation by dialysis against PBS (8 × 1 L, 24 h; Slide-A-Lyzer Dialysis Cassettes, MWCO 3.5 kDa). The formulations were stored at 4 °C until use. The average size (hydrodynamic diameter) distribution (SD) and polydispersity index (PDI) were measured by dynamic light scattering and calculated from three independent liposomal preparations (DLS, Zetasizer Nano ZS, Malvern Instruments Ltd., Malvern, UK). The PDI is a measure of the degree of uniformity of the size distribution of the liposome suspension.

### 2.4. Human Tissue

Term placentas (32–40 weeks gestation) were obtained from pregnancies within 30 min of vaginal (FGR cohort) or caesarean delivery (healthy pregnancy cohort). Healthy pregnancy was defined as an individualized birth weight ratio (IBR) between the 10th to 90th centiles and FGR was defined as an IBR below the 3rd centile. Patients with renal disease, premature rupture of membranes, fetal anomalies or chromosomal abnormalities were excluded from the study. Patient demographic details are shown in Table 1. Human placental tissue collection was conducted according to the guidelines of the Declaration of Helsinki and approved by the NHS Local Research Ethics Committee (LREC) and the University Research Ethics Committee (UREC) at The University of Manchester, UK (approval code: 18/WA/0356, approval date: 1 October 2017); written informed consent was obtained from all patients. Villous tissue was randomly sampled from center, middle and edge areas of the placenta and placed in serum-free medium (1:1 Dulbecco’s modified Eagle Medium (DMEM) and Ham’s F12 (Lonza Biosciences, London, UK) supplemented with penicillin (100 IU/mL), streptomycin (100 µg/mL), glutamine (2 mM) and amphotericin B (2.5 µg/mL) (Invitrogen, Inchinnan, UK)) prior to tissue culture experiments [7,18].

### 2.5. Explant Culture

One cm^3^ placental villous tissue biopsies (with the chorionic plate and decidua removed) were sampled at random and dissected into ~ 2–3 mm^3^ explants. To remove blood cells, the villous tissue was washed in fresh serum-free medium (as above). Three 2–3 mm^3^ explants were then cultured at 37 °C (21% O_2_, 5% CO_2_) in individual wells of a 12-well Costar Netwell plate (Corning Life Sciences, Corning, NY, USA) containing culture medium (CMRL-1066 (100 mL/L), NaHCO_3_ (2.2 mg/mL), streptomycin sulphate (100 μg/mL), penicillin G (100 IU/mL), insulin (1 μg/mL), retinol acetate (1 μg/mL), L-glutamine (100 μg/mL) and 5% fetal bovine serum (pH 7.2, Invitrogen Corporation, Paisley, UK)). Culture medium was replaced daily up to and including day 5 and sampled on days 2, 5 and 7. Medium was stored at −20 °C for further biochemical analyses. Explants were treated on day 5 of culture with either vehicle (PBS), free EGF (50 or 100 ng/mL), non-targeted liposomes containing EGF (50 ng/mL; 2.5% *v/v*) or targeted liposomes containing EGF (50 ng/mL; 2.5% *v/v*), and maintained for 48 h until day 7. The EGF concentrations of 50 and 100 ng/mL were chosen based on previously published data examining the effects of EGF on term placental explants [18,19]. A single concentration of 100 ng/mL EGF was used in healthy placental explants, whereas two concentrations of 50 and 100 ng/mL were used in FGR explants to assess any possible concentration-dependent responses. Explants were fixed overnight in 4% neutral buffered formalin (NBF; 4% *v/v*, pH 7.4) at 4 °C. Fixed tissue was washed 3 times with PBS and embedded in paraffin wax.

### 2.6. Immunohistochemistry

Tissue sections (5 µm) were cut and transferred onto poly-L-lysine-coated slides. After dewaxing in Histoclear and rehydration in alcohol, antigen retrieval was performed by microwaving (800 W) the slides for 10 min in sodium citrate buffer (0.01 M; pH 6.0). After cooling, slides were washed in distilled water and incubated with 3% (*v/v*) hydrogen peroxide for 10 min to block endogenous peroxidase activity. Slides were incubated with the non-immune block (10% goat serum and 2% human serum in 0.1% TBST (TBS-Tween-20)) for 30 min at room temperature, before application of primary antibodies specific for Ki67 (mouse monoclonal: 0.16 μg/mL; Dako, Ely, UK) or M30 (mouse monoclonal: 0.17 μg/mL; Roche, London, UK), as previously described [23]. Sections were incubated overnight at 4 °C, washed (5 min in TBS; 2 × 5 min in TBS-Tween (0.6% *v/v*); 5 min in TBS), then the secondary antibodies were applied (horse- or goat-anti-mouse IgG, 1:200; Dako). Slides were incubated for 30 min at room temperature, washed again (5 min in TBS; 2 × 5 min in TBS-Tween (0.6% *v/v*); 5 min in TBS), then incubated with avidin peroxidase (5 μg/mL, Sigma-Aldrich, Gillingham, UK) for 30 min at room temperature. After further TBS washes as described above, tissue sections were incubated with 0.05% (*w/v*) 3,3′-diaminobenzidine (DAB) and 0.015% (*v/v*) hydrogen peroxide (Sigma-Aldrich, Gillingham, UK), and color development was monitored under a light microscope. Slides were washed in distilled water, counterstained with hematoxylin, dehydrated in alcohol and Histoclear and mounted in DPX (Sigma-Aldrich, Gillingham, UK). Negative controls were performed by substitution of primary antibody with non-immune rabbit IgG. Six fields of view per section were obtained on a Leica DMRB microscope at 20× magnification. Histoquest (TissueGnostics, Vienna, Austria) quantification analyses calculated the total number of DAB-positive nuclei expressed as a percentage of total hematoxylin-stained nuclei (Ki67), or the total area of DAB-positive staining expressed as a percentage of total tissue area (M30).

### 2.7. Biochemical Assays

hCG and lactate dehydrogenase (LDH) release into culture medium was quantified by ELISA (Immunodiagnostics, Sunderland, UK) or the cytotoxicity detection colorimetric assay (Sigma Aldrich, Gillingham, UK) respectively, as per manufacturers’ instructions [18]. hCG and LDH were normalized to explant protein content using a standard Bio-Rad protein assay (Bio-Rad, Hertfordshire, UK).

### 2.8. Measurement of System a Transporter Activity

System A transporter activity was measured as previously described [24]. On day 7, placental explants were removed from Costar Netwells and placed in preconditioning media (a 1:1 ratio of Dulbecco’s Modified Eagle’s Medium/Nutrient Mixture F-12 Ham (Sigma Aldrich, Gillingham, UK) and Na^+^-containing Tyrode’s buffer) then incubated for 30, 60 or 90 min in either Na^+^-containing or Na^+^-free Tyrode’s buffer, which contained 0.5 µCi/mL of ^14^C-Methylaminoisobutyric acid (MeAIB; ~8.5 µM) at 37 °C (21% O_2_). Explants were removed and immersed in dH_2_O for a minimum of 18 h at room temperature. The radioactivity of the dH_2_O lysate, reflecting the uptake of ^14^C MeAIB into the explants, was measured using a β-counter and converted to moles using the specific activity of ^14^C MeAIB. Na-dependent system A transporter activity was deduced by subtracting the uptake of ^14^C MeAIB in Na^+^-free Tyrode’s buffer from the uptake in Na^+^-containing (control) Tyrode’s buffer. System A activity was normalized to explant protein content using a standard Bio-Rad protein assay (Bio-Rad, Hertfordshire, UK).

### 2.9. Phosphokinase Array

Placental explant cultures were established as previously described [7,18]. On day 5 of culture, explants were treated with either PBS (control), EGF (100 ng/mL), non-targeted liposomes containing EGF (50 ng/mL) or targeted liposomes containing EGF (100 ng/mL) and incubated for a further 24 h. Explants were harvested, transferred to RIPA buffer (Sigma Aldrich, Gillingham, UK; containing 1:100 phosphatase inhibitor cocktail, 1:100 protease inhibitor cocktail) and homogenized in a Bullet Blender (Next Advance, Troy, NY, USA) at speed 8 for 10 min at 4 °C. The resulting lysates were spun at 2000× *g* for 10 min, and supernatants were collected and transferred to a fresh Eppendorf tube. A BCA assay (Thermo Fisher Scientific, Loughborough, UK) was performed to quantify the protein concentration. A Proteome Profiler Human Phospho-Kinase array kit (R&D Systems) was used as per the manufacturer’s instructions. Briefly, supernatants were incubated with nitrocellulose membranes impregnated with capture antibodies. After washing, captured proteins were detected with biotinylated antibodies and visualized using chemiluminescent detection reagents. Images were captured on a Gel Doc XR+ (Biorad laboratories Ltd., London, UK), and signal intensity was quantified using ImageJ and normalized to internal reference spots.

The protein expression data from the array were analyzed using R studio (1.3.1056). The 13 proteins which were differentially expressed, and clustered together in treatments vs. control, were subject to STRING analyses (https://string-db.org/, (accessed on 25 October 2019)) using the multiple proteins options, to detect predicted protein interactions and signaling pathways. The genes associated with the 13 differentially expressed proteins were selected and entered in PANTHER (http://www.pantherdb.org, (accessed on 25 October 2019)) to identify molecular functions affected by the treatments.

### 2.10. Statistical Analyses

Unless otherwise stated, all statistical analyses were performed using GraphPad Prism Software V8.1 (San Diego, CA, USA). hCG and LDH percent change in release between day 5 and day 7 of culture was analyzed using a Wilcoxon test to compare values to a hypothetical value of zero % change. Data are presented as median. Ki67 and M30 immunostaining were analyzed using a Kruskal–Wallis test. Data are presented as median. System A amino acid transporter activity was analyzed by linear regression to identify differences in Na^+^-dependent and Na^+^-independent uptake of radioisotope between treatments. Ninety-minute Na^+^-dependent uptake data are also presented as a percentage of matched placenta and analyzed using a Wilcoxon test. Demographic data were analyzed by the Holm–Sidak test. A *p*-value < 0.05 was considered significant.

## 3. Results

### 3.1. CCGPSARAPC Peptide Localizes to Human Placenta

Prior to in vivo screening, a library of T7 bacteriophages was incubated with homogenized mouse placental tissue obtained at embryonic (E) days 13.5 and 14.5 of pregnancy, where E19 is full-term in this mouse model. The library contained 10^8^ individual viral clones, which each expressed a unique random 9mer cyclic peptide sequence, in the form CXXXXXXXC, on their surface. A single virus displayed multiple copies of the peptide, but only one sequence was represented per viral particle. As the library became preferentially enriched for placental-binding peptides, the diversity of the phage library collapsed (Figure 1A). After three rounds of enrichment, the purified viral pool was intravenously injected into pregnant dams for the in vivo screening (Figure 1B). Fifty phage clones from the fourth round of in vivo screening were sequenced, in which the CGPSARAPC surface peptide was highly represented.

Synthetic versions of this peptide were created, with the addition of a second N-terminal cysteine residue to facilitate fluorophore and liposomal conjugation: CCGPSARAPC (GPS). This sequence was labeled with either 5(6)-carboxyfluorescein (FAM) or 5-carboxytetramethylrhodamine (TAMRA) to visualize the location of peptide homing in both mouse (Figure 1C–J) and human tissues (Figure 1K,L). FAM-GPS was intravenously injected into pregnant mice, where it selectively localized to the junctional zone of the mouse placenta (Figure 1C,D). Analysis of maternal tissues showed no binding of FAM-GPS to the lung, brain or heart (Figure 1G,I,J). As previously reported [7], small discrete areas of fluorescence were observed in clearance organs, such as the liver and spleen (Figure 1E,F,H). Free TAMRA-GPS peptide was incubated ex vivo with explants of human term placental tissue and was found to localize to the outermost syncytiotrophoblast layer (Figure 1K).

### 3.2. Liposomal Preparation and Characterization

To create nanoparticles for targeted delivery of payloads to the placenta, liposomes were prepared from cholesterol, DSPC, DSPE and PEG, and TAMRA-GPS was covalently conjugated to maleimide groups on the liposomal surface. Four formulations were prepared: liposomes containing carboxyfluorescein (CFS), with and without GPS-decoration, to assess liposomal uptake and payload release, and liposomes containing EGF, with and without GPS-decoration, to assess modification of placental function. The mean diameters of non-targeted and targeted CFS liposomes were 206.5 ± 7.0 and 186 ± 2.6 nm, respectively. The polydispersity indices for each formulation were 0.360 ± 0.19 and 0.234 ± 0.06. The mean diameters of the non-targeted and targeted EGF liposomes were 182 ± 4.2 and 201 ± 3.3 nm, respectively. The polydispersity indices were 0.038 ± 0.04 and 0.11 ± 0.07. Incubation of term placental tissue explants with GPS-decorated liposomes led to placental-specific binding and uptake, particularly in the outer syncytiotrophoblast layer (Figure 1L).

### 3.3. GPS-Liposomes Do Not Alter the Basal Rate of hCG Secretion, Cell Turnover or System a Transporter Activity

Having confirmed binding and uptake, we assessed whether the liposomes altered any parameters of normal placental growth and function. Term placental explants from healthy pregnancies were cultured with non-targeted liposomes containing CFS or GPS-decorated liposomes containing CFS (targeted liposomes + CFS) for 48 h. Exposure to these formulations had no significant effect on the basal rate of hCG secretion (Figure 2A), the percentage of cells in cycle (Figure 2B), the percentage of apoptotic cells (Figure 2C) or the basal system A amino acid transporter activity (Figure 2D).

### 3.4. Delivery of EGF Does Not Alter hCG Secretion, LDH Release or Cell Turnover

To determine whether EGF could be used to improve different parameters of placental growth and viability in healthy and FGR placentas, non-targeted liposomes containing EGF, targeted liposomes containing EGF and free EGF were incubated with human placental explants from day 5 to day 7 of explant culture (48 h total). As has previously been reported [25], we observed a reduction in hCG secretion from healthy explants between day 5 and day 7 of culture (Figure 3A), which was also evident in explants from FGR pregnancies (Figure 3C). However, the decrease in hCG production between day 5 and day 7 was less pronounced in explants from healthy pregnancies incubated with targeted liposomes containing EGF (Figure 3A). This effect was not observed in explants from FGR pregnancies (Figure 3C). Similarly, no significant change in hCG secretion was noted in explants treated with free EGF, or non-targeted liposomes containing EGF, from either healthy or FGR pregnancies (Figure 3A,C). No concentration-dependent effects of free EGF were observed for FGR explants, so a concentration of 100 ng/mL was used for the majority of subsequent experiments.

Release of LDH from placental explants was also measured before (on day 5) and after (on day 7) EGF treatment. Free EGF, non-targeted liposomes containing EGF and targeted liposomes containing EGF had no effect on basal levels of LDH release in placental explants from either healthy (Figure 3B) or FGR (Figure 3D) pregnancies. Furthermore, there was no effect of free EGF, non-targeted liposomes containing EGF or targeted liposomes containing EGF on the percentage of apoptotic cells, or cells in cycle in explants from either healthy (Figure 4A,B) or FGR placentas (Figure 4C,D).

### 3.5. EGF Significantly Increases System a Transporter Activity in Human Placental Explants

The Na^+^-dependent uptake of ^14^C-MeAIB, a non-metabolizable substrate of the system A transporter, was measured in human placental explants from healthy and FGR pregnancies (Figure 5). Uptake of ^14^C-MeAIB was linear over 30–90 min in explants from both healthy (Figure 5A) and FGR (Figure 5B) placentas. Incubation of placental explants with free EGF (50 or 100 ng/mL) for 48 h led to a significant increase in system A transporter activity in healthy (Figure 5A, *p* < 0.05) but not FGR explants (Figure 5B). Incubation of placental explants with targeted liposomes containing EGF also led to a significant increase in system A transporter activity in explants from healthy (Figure 5A) but not FGR placentas (Figure 5B). These increases were most apparent after 90 min, when displayed as a percentage of the activity observed in matched control explants (Figure 5C,D). In almost all cases, system A transporter activity was increased in response to free, non-targeted and targeted EGF in healthy placental explants (Figure 5C), as the vast majority of data points lie above the 100% control line. However, FGR explants could be divided into responders and non-responders, with 40–50% of data points falling below the 100% control line (Figure 5D), suggesting that some, but not all, placentas had the capacity for functional rescue. Na^+^-independent uptake of ^14^C-MeAIB was also measured to ensure that treatments did not alter the integrity of placental tissue. Na^+^-independent uptake of ^14^C-MeAIB was not different between healthy and FGR placentas, and was not affected by treatment (data not shown).

### 3.6. Phosphorylated Kinase Activity in Placental Explants in Response to EGF Delivery

To confirm that liposomally encapsulated EGF could activate the EGFR, placental explants from normal pregnancies were treated with free EGF, non-targeted EGF or targeted EGF. A phosphokinase array was used to assess the protein kinases activated in response to each treatment. The extent of kinase phosphorylation is shown in the heat map in Figure 6A. These data confirm that liposomally delivered EGF can still activate EGF-receptor signaling, with delivery in targeted liposomes more accurately recapitulating the phosphokinase signaling observed following treatment with free EGF. Hierarchal clustering analyses of the phosphorylation data using R identified 13 phosphorylated protein kinases which were differentially expressed in response to all EGF treatments vs. control (red box, Figure 6A). Functional protein analyses of these 13 differentially regulated protein kinases demonstrated the ErbB signaling network to be highly associated with EGF treatment (STRING analyses, Figure 6B). PANTHER analyses identified catalytic activity (56.2%), binding (42.1%) and transcription regulation (5.3%) as the molecular functions most associated with, and affected by, the EGF treatments (Figure 6C).

## 4. Discussion

In this study, we demonstrate that liposomes decorated with a peptide that selectively homes to the placental surface can be used to deliver EGF to human placental explants, enhancing amino acid transport and ameliorating the reduction in hCG secretion observed in culture. Furthermore, we demonstrate that liposomal delivery of EGF closely recapitulates the tyrosine kinase signaling cascade observed when explants are exposed to free EGF, suggesting that this delivery method does not impair its ability to activate the EGF receptor.

The rationale for using targeted drug delivery strategies in pregnancy is similar to that of cancer: controlling the location of payload delivery. In cancer therapy, the aim is to kill the tumor cells, and selecting a nanoparticle carrier with cytotoxic properties further enhances the efficacy of the treatment. However, in pregnancy, the aim is to deliver therapies that enhance placental growth, rather than induce a cell death or damage response. Many of the interventions that enhance placental cell growth, survival and function in vitro cannot be administered systemically, due to the risk of cell overgrowth or tumor formation in other organs, or causing developmental abnormalities in the fetus. Thus, placental-specific delivery is critical for patient safety. In addition, the nanoparticle carrier must not directly or indirectly induce toxicity or teratogenicity. Numerous studies have shown that a variety of inorganic nanoparticles induce cell death and developmental abnormalities in the placentas and fetuses of mice, rats and rabbits, and cause direct damage to human placental cells and tissue explants treated ex vivo [26]. It has also been hypothesized that exposure to inorganic nanoparticles can lead to damage signals that cross the placental barrier and elicit neuronal toxicity in the fetus [27]. Thus, using an organic nanoparticle formulation for drug delivery in pregnancy is another critical requirement. Liposomes represent a non-toxic, biodegradable formulation which already has FDA approval for several indications. In this study, we have confirmed that our liposomal formulation does not alter basal levels of hCG secretion (a marker of syncytiotrophoblast differentiation) or LDH release (a marker of tissue damage), does not induce placental cell death or alter cell turnover and does not interfere with amino acid transport. Although we observed some variability in individual placental characteristics in our study, as is common with any biological dataset, all reported values fell well within the ranges reported by our group and others [7,18,28]. Moreover, the consistent lack of response to control formulations is entirely consistent with our previous work [7,29].

A number of previous studies have reported that EGF treatment of human placental cell lines and isolated primary placental cells can enhance cell motility and invasion [30] and protect against hypoxia-induced apoptosis [31]. In the first trimester of pregnancy, EGF increases proliferation [32] and hCG secretion from isolated cytotrophoblast cells [32,33]; at term, the spontaneous differentiation of isolated cytotrophoblasts into syncytiotrophoblasts can be enhanced by EGF [19,34]. However, there is a much smaller body of work documenting the effects of EGF in a tissue explant model. EGF reportedly protects against apoptosis induced by inflammatory cytokines and oxidative stress in term placental explants [18,19,20]; however, we observed that EGF did not alter the basal rate of apoptosis in either healthy or FGR placentas. We did observe that all formulations of EGF were able to significantly enhance nutrient transport capacity via the system A transporter, the principal means by which neutral amino acids are transferred to the fetal circulation. In contrast, we failed to observe enhancement of system A activity in FGR placentas. It is critical to remember that FGR is a multifactorial disease [3], and the underlying placental dysfunction may be caused by abnormalities in placental structure, cell turnover, hormone secretion, nutrient transfer capacity and/or vascular function. Thus, if FGR was caused by vascular defects and impaired uteroplacental perfusion in one patient, but by reduced trophoblast proliferation and a smaller placenta in another, we might hypothesize that EGF treatment would improve the underlying pathology in the latter patient but not the former. Alternatively, in the case of nutrient transport capacity, the concentration of EGF administered or the single acute exposure may not have been sufficient to induce rescue, because the placental pathology was too severe to achieve significant improvement, and/or the system A transporters were already operating at maximum capacity. Indeed, we observed that some, but not all, FGR explants responded to treatment, suggesting that our EGF treatment regimen was sufficient to improve capacity in some samples. The median birthweight centile of the FGR pregnancies we investigated was 0.2; clinically, any baby falling below the third weight centile is considered growth-restricted. Thus, the placentas we studied likely exhibited significant pathology and may not have been amenable to rescue at this stage of gestation.

The use of liposomes for advancing drug delivery in pregnancy is a relatively recent phenomenon. Studies have explored how the composition and size of non-targeted liposomes regulates transplacental passage using ex vivo, dual-perfused human placental cotyledons, with the aim of enhancing fetal thyroxine delivery [35] or retaining warfarin [36] or valproic acid [37] within the maternal circulation. In animal models, targeted liposomes have been used for delivery of indomethacin to the uterus, for the treatment of pre-term labor [38,39]. Similarly, targeted liposomes have been used to deliver insulin-like growth factor-I and -II [7], SE175 [29] and gentamicin [40] to the placenta, to increase trophoblast proliferation, improve uteroplacental blood flow and enhance placental liposome uptake, respectively. There are few reported studies of liposomal uptake and drug delivery in human placental explants, other than our own. Dual-labeled, non-targeted liposomes accumulated in the syncytiotrophoblast of human term placental explants cultured in vitro at a similar rate and extent to our own untargeted formulations [41]. Non-targeted liposomes containing siRNA also selectively accumulated in the syncytiotrophoblast [42]. More thorough studies of liposomal interactions with the human placenta are certainly warranted.

Treatment of human placental explants with free EGF leads to phosphorylation of the EGFR and activation of a protein kinase signaling cascade [43]. To confirm that liposomally encapsulated EGF could enter the syncytiotrophoblast, exiting the liposome and interacting with the EGF receptor to initiate signaling, we assessed kinase activity downstream of the EGF receptor. As it was unclear whether encapsulated EGF would lead to classical activation of the EGF signaling pathway, given the unconventional mode of delivery, a non-specific phosphokinase array was used. The array data show that encapsulation of EGF inside liposomes did not impair its ability to signal through the EGF receptor, with targeted delivery more accurately reflecting the signaling cascade induced by free EGF. Undecorated liposomes are believed to be internalized via clathrin- or caveolin-mediated endocytosis, eventually reaching the late endosomal or lysosomal compartments [44], where cargo degradation may occur. Addition of ligands to the liposomal surface can shift the mechanism of uptake to receptor-mediated endocytosis, in which lysosomal degradation may be avoided [21,44]. Thus, the unique kinase activation profile associated with delivery of EGF in non-targeted liposomes may reflect temporal and/or spatial differences in its interactions with the EGFR. Moreover, identification of ErbB signaling in response to EGF treatment via STRING analysis provides additional evidence of EGFR activation. EGFR is a member of the ErbB family of receptor tyrosine kinases, and each receptor can dimerize with and activate other ErbB family members upon ligand binding [45]. However, we acknowledge that further work is required to more comprehensively assess the exact signaling pathways activated by each of our liposomal formulations and the specific functional effects of EGF stimulation.

## 5. Conclusions

This study has identified the GPS sequence as a new homing peptide for placental-specific drug delivery and has demonstrated that GPS-decorated liposomes represent a safe and useful tool for the development of new targeted therapies for placental dysfunction. Whilst free and liposomally encapsulated EGF did not enhance placental function to the extent which we predicted, it did increase nutrient transport capacity in most healthy explants and in approximately 50% of FGR explants, and still represents a promising therapeutic candidate worthy of further investigation.

## Figures and Tables

**Figure 1 pharmaceutics-13-01778-f001:**
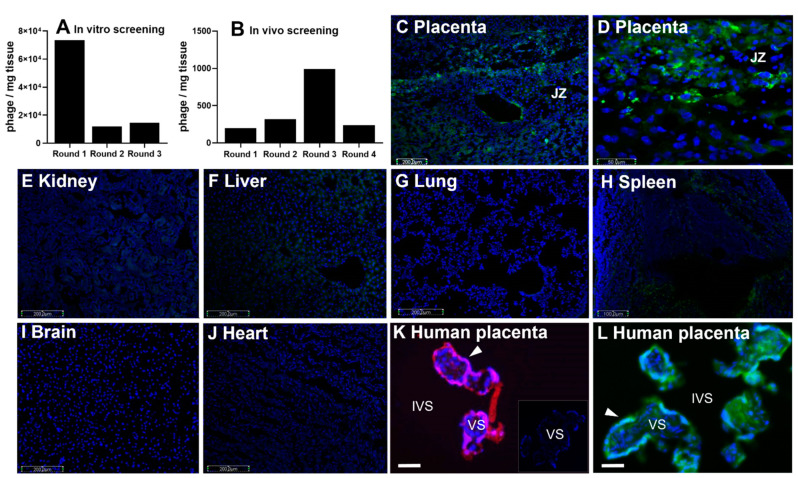
Identification and localization of placental-homing peptide CGPSARAPC (GPS). (**A**) Mouse placental tissue was incubated with a T7 bacteriophage CX7C library and bound phages were tittered, amplified and purified. Phage titer (pfu/mg tissue) was measured using a plaque-forming assay (median ± SEM; *n* = 3). (**B**) After 3 rounds of ex vivo selection, the resulting phage pool was injected into the tail vein of a pregnant mouse. Placentas were harvested and bound phage pools were tittered, amplified and purified. Four rounds of in vivo screening were performed, and phage titer is expressed as pfu/mg tissue (median ± SEM; *n* = 3). (**C**–**J**) CCGPSARAPC labeled with 5(6)-carboxyfluorescein (CFS; 200 μg) was injected into the tail vein of pregnant mice (*n* = 4; E0.5–E17.5). Following cardiac perfusion to remove unbound peptide, placentas (**C**,**D**) and maternal organs (**E**–**J**) were harvested and assessed by fluorescence microscopy. (**K**,**L**) Human placental explants were incubated with TAMRA-CCGPSARAPC (0.27 µmol/L; 3 h; *n* = 3) (**K**), no peptide (inset in (**K**)) or liposomes decorated with CFS-CCGPSARAPC (24 h; *n* = 3). Peptide binding was assessed by fluorescence microscopy, and representative images are shown. CFS-labeled peptides, green; TAMRA-labeled peptides, red; DAPI (nuclei), blue. JZ = Junctional zone; VS = villous stroma; IVS = intervillous space. Arrow denotes syncytiotrophoblast. Scale bar = 50–200 µm.

**Figure 2 pharmaceutics-13-01778-f002:**
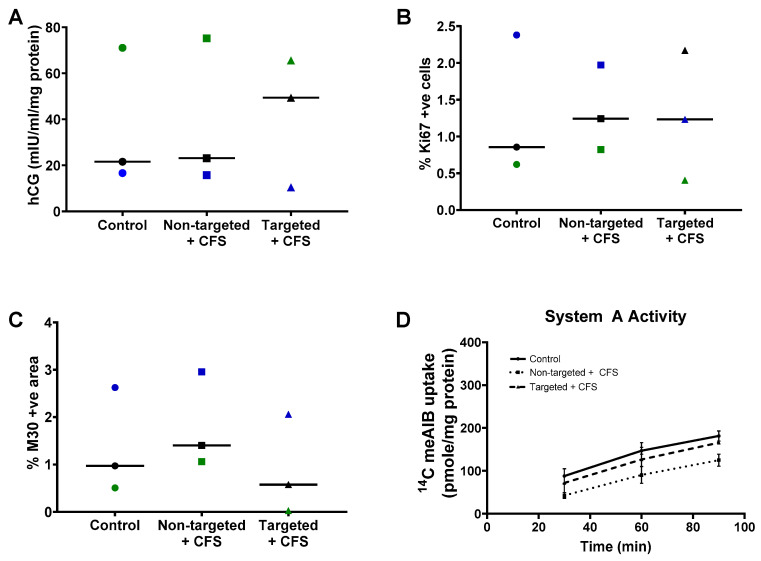
Liposomal formulations do not alter normal placental function. Healthy third-trimester human placental explants were cultured for 7 days and treated with either 10 µL PBS (control), undecorated liposomes encapsulating carboxyfluorescein cargo (Non-targeted + CFS) or GPS-decorated liposomes encapsulating carboxyfluorescein cargo (Targeted + CFS) for 48 h on days 5–7. hCG secretion; *n* = 3 (**A**), cells positive for Ki67; *n* = 3 (**B**), M30-positive area; *n* = 3 (**C**), and System A amino acid transporter activity; *n* = 2–3 (**D**), measured 48 h after treatment (day 7). Matched samples are shown in blue, green or black in (**A**−**C**). Median values are shown. Data not significantly different. hCG = human chorionic gonadotropin.

**Figure 3 pharmaceutics-13-01778-f003:**
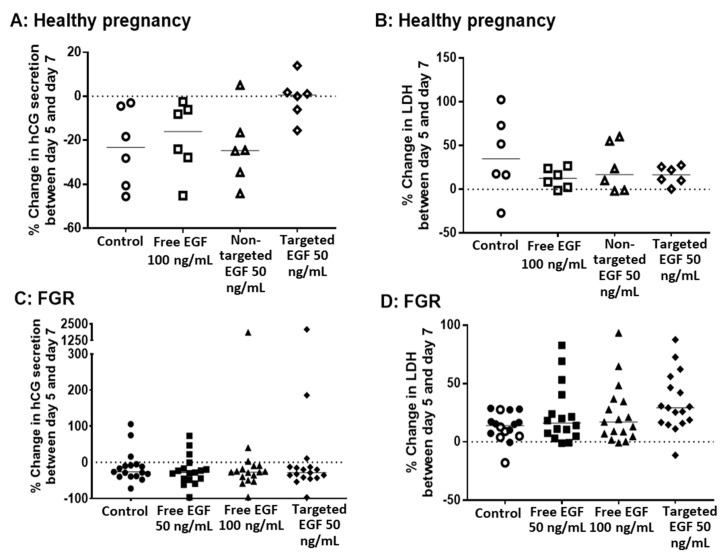
Targeted delivery of EGF does not alter placental secretion of hCG or release of LDH in healthy pregnancy or FGR. Healthy (**A**,**B**) and FGR (**C**,**D**) third-trimester human placental explants were cultured for 7 days and treated with either 10 µL PBS (control), free EGF (50 or 100 ng/mL), undecorated liposomes encapsulating 50 ng/mL EGF (non-targeted) or GPS-decorated liposomes encapsulating 50 ng/mL EGF (targeted) for 48 h on days 5–7. Percent change in hCG secretion (**A**,**C**) and LDH release (**B**,**D**) between day 5 and day 7, as measured by ELISA. *n* = 6 healthy pregnancy, *n* = 17 FGR. Data not significantly different. EGF = epidermal growth factor, FGR = fetal growth restriction, hCG = human chorionic gonadotropin, LDH = lactate dehydrogenase.

**Figure 4 pharmaceutics-13-01778-f004:**
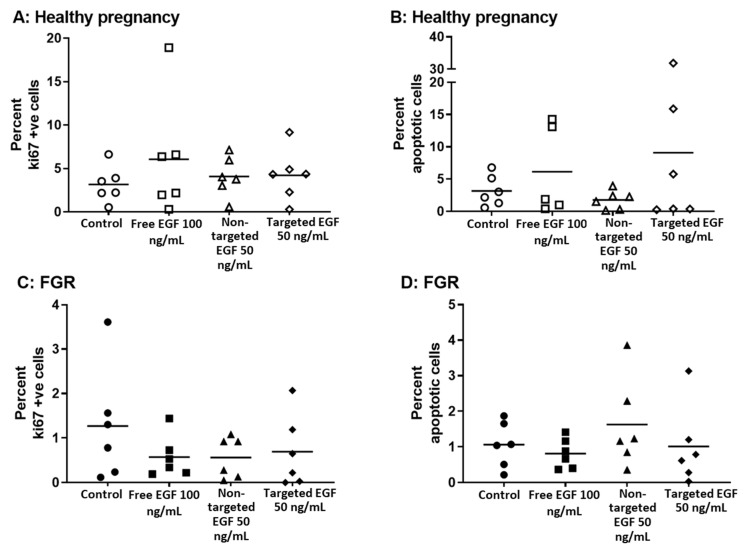
Targeted delivery of EGF does not alter the percentage of placental cells in cycle or undergoing apoptosis in healthy pregnancy or FGR. Healthy (**A**,**B**) and FGR (**C**,**D**) third-trimester human placental explants were cultured for 7 days and treated with either 10 µL PBS (control), free EGF (100 ng/mL), undecorated liposomes encapsulating 50 ng/mL EGF (non-targeted) or GPS-decorated liposomes encapsulating 50 ng/mL EGF (targeted) for 48 h on days 5–7. Percent of M30-positive (**B**,**D**) and Ki67-positive (**A**,**C**) apoptotic and cells in cycle, respectively. Mean, *n* = 5–6 placentas. Data not significantly different. EGF = epidermal growth factor, FGR = fetal growth restriction.

**Figure 5 pharmaceutics-13-01778-f005:**
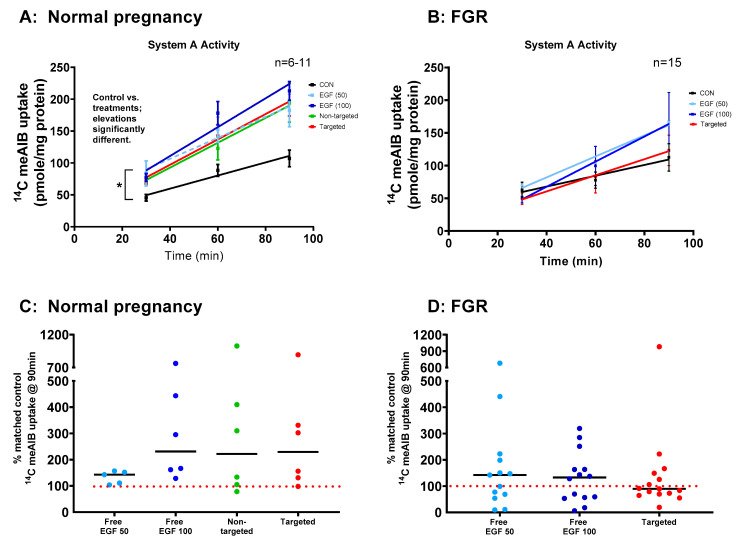
Targeted delivery of EGF increases System A amino acid transport activity in human placental explants. Third-trimester explants from healthy (**A**) and FGR (**B**) pregnancies were cultured for 7 days and treated with either 10 µL PBS (control), free EGF (50 or 100 ng/mL), undecorated liposomes encapsulating 50 ng/mL EGF (non-targeted) or GPS-decorated liposomes, encapsulating 50 ng/mL EGF (targeted) for 48 h on days 5–7. Na^+^-dependent uptake of ^14^C MeAIB (**A**,**B**) (System A activity) measured at 30, 60 and 90 min. Mean ± SEM. *n* = 6–15 placentas. Asterisk donates the linear regression elevations are significantly different (*p* < 0.05) in (**A**) with individual post hoc analyses; control vs. EGF 50 ng/mL *p* = 0.02, control vs. EGF 100 ng/mL *p* < 0.0001, control vs. non-targeted *p* = 0.04, control vs. targeted *p* = 0.03. (**C**,**D**) Percentage of Na^+^-dependent uptake of ^14^C MeAIB compared to levels in matched control explants at 90 min. Median, *n* = 6–15 placentas. EGF = epidermal growth factor, FGR = fetal growth restriction, MeAIB = 2-Methylaminoisobutyric acid.

**Figure 6 pharmaceutics-13-01778-f006:**
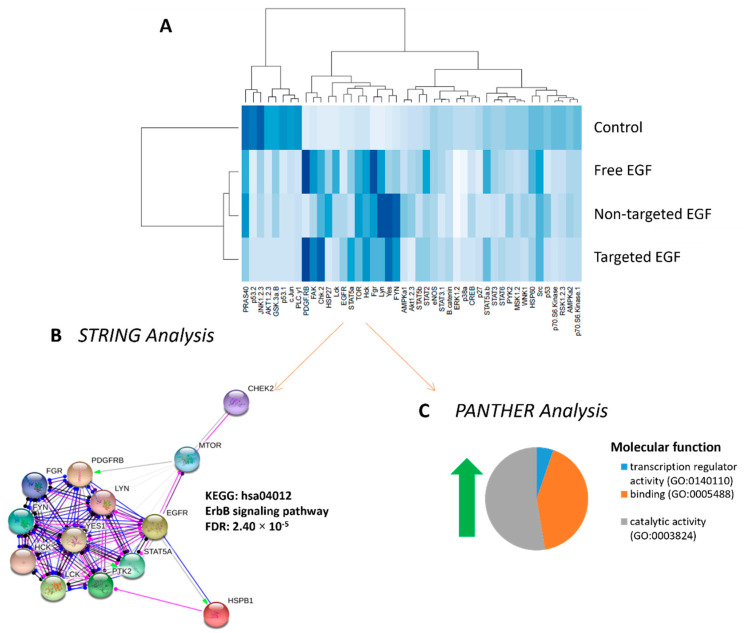
Phosphorylation of downstream protein kinases in response to EGF delivery. Tissue lysates were prepared from human placental explants from healthy pregnancies which had been treated with PBS (control), free EGF (100 ng/mL), undecorated liposomes encapsulating 50 ng/mL EGF (non-targeted) or GPS-decorated liposomes encapsulating 50 ng/mL EGF (targeted) for up to 3 h. Phosphorylation of kinase signaling molecules was assessed in pooled lysates using a phosphokinase array. Hierarchical cluster analysis of phosphorylation intensity data grouped similarly phosphorylated proteins together, and darker blue colors demonstrate more highly phosphorylated targets (**A**). STRING analyses of kinases with increased phosphorylation in response to EGF was strongly associated with activation of the ErbB signaling pathway (**B**). PANTHER analysis of kinases with increased phosphorylation in response to EGF identified catalytic activity, transcription regulator activity and binding as the three most common molecular functions altered in response to treatment (**C**). EGF = epidermal growth factor., p38a = p38 mitogen activated protein kinases a, ERK = extracellular signal-regulated kinase, JNK = c-Jun N-terminal kinase, GSK = glycogen synthase kinase, p53 = tumor protein P53, EGFR = epidermal growth factor receptor, MSK = mitogen and stress induced kinase, AMPK = AMP-activated protein kinase, AKT = protein kinase B, TOR = target of rapamycin, CREB = cAMP response element-binding protein, HSP = heat shock protein, p70 S6 kinase = ribosomal protein S6 kinase beta-1, Src = proto-oncogene tyrosine-protein kinase, STAT = signal transducer and activator of transcription, RSK = ribosomal S6 kinase. eNOS = endothelial nitric oxide synthase, p27 = cyclin-dependent kinase inhibitor 1B, PLC-γ1 = phospholipase c, gamma 1, Hck = Hematopoietic cell kinase, Chk = checkpoint kinase, FAK = focal adhesion kinase, PDGF R = platelet derived growth factor receptor, WNK = with no lysine, PYK = pyruvate kinase, PRAS = proline-rich AKT substrate, PBS = phosphate buffered saline.

**Table 1 pharmaceutics-13-01778-t001:** Patient demographics.

Demographic	Normal Pregnancy (*n* = 6)	FGR (*n* = 15)	*p*-Value
Maternal age (years)	35 (31–37)	29 (24–48)	NS
Parity	2 (1–2)	0 (0–2)	NS
Smoker	0/6	4/15	
BMI	25 (19–29)	24 (19–47)	NS
Gestation (days)	273 (272–280)	250 (209–268)	*p* = 0.002
Male sex	6/6	3/15	
Individualized birthweight centile	52 (14–80)	0.2 (0–3)	*p* < 0.0001

Median (range). Statistical significance determined by Holm–Sidak test. NS = not significant, FGR = fetal growth restriction, BMI = body mass index.

## Data Availability

Data is available on request.

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
