# Peer review of "Targeted Delivery of Epidermal Growth Factor to the Human Placenta to Treat Fetal Growth Restriction"

_pharmaceutics, 2021, doi:10.3390/pharmaceutics13111778_

Round 1

Reviewer 1 Report

The manuscript outlines a liposome based targeted delivery system for placental cells. The work is interesting and definitely important for the field as specific delivery of drugs or interventions to the placenta is difficult. The manuscript is well written, but the following comments need to be addressed.

  1. Can the authors elaborate on the “polydispersity index” value and how it was calculated?
  2. For figure 2 assessing effects of targeted and non taregtted liposome formulations, the authors need to add more n numbers. The data has high variability which casts a shadow on their interpretations
  3. Did the authors assess what’s the rate/extent of uptake of the liposomal particles?
  4. In addition, maybe I missed this, but are the samples in figure 2 matched? i.e. sample placenta across the three treatment groups?
  5. N numbers for Figure 3, discussing effects on hCG and LDH secretion are missing.
  6. Figure 3 shows two concentrations of free EGF only for FGR explants, but the authors do not really discuss the dosage or the reason for two dosages just for FGR in the text.
  7. To follow up on point 5 above, figure 4 C & D data sets only show one free FGR column. What concentration was this and why was it chosen?
  8. In figure 5D, the authors suggest that FGR explants could be divided into responders and non-responders. What is the basis/logic for this conclusion? The controls in 5C also appear to split into two groups, so do the samples in Figure 4B, 4C and 4D. How do the authors explain this split in their samples? The high variability could explain the absence of significant changes in the samples due to EGF, but that cast a doubt upon the reliability of the targeted system.
  9. Lines 455-456; term cytotrophoblast spontaneously differentiate to syncytiotrophoblast in culture and do not need EGF stimulation.
  10. The authors need a better and specific parameter for evaluating the effects EGF targeted delivery. The kinase array is not specific for EGF and assessing specific downstream activity might help support their conclusion.
  11. As the authors indicate the effects of EGF in explants has not been completely explored. Their results on ki67 staining- for proliferating cells and % apoptotic cells are at best confusing – for all three treatment modes. While the authors provide interesting data about specific delivery of the particles (and I am impressed with the immense work that went to it) their EGF targeting data fails to deliver the impact of their technology. Narrowing down to specific effects of EGF and evaluating those will significantly help improve the manuscript.

Author Response

Response to reviewers

Please note that at the request of the editor some changes have been made to the methods, as the original text was deemed too similar to that of previous publications.

Reviewer 1:

Can the authors elaborate on the “polydispersity index” value and how it was calculated?

The polydispersity index (PDI) is a measure of the degree of uniformity of a size distribution of particles. A liposome formulation containing nanoparticles of similar diameter is desirable, and is termed monodisperse. A formulation containing nanoparticles with a wide range of diameters is termed polydisperse. The PDI is calculated from the square of the standard deviation, divided by the mean particle diameter. It is measured by dynamic light scattering using a Zetasizer Nano ZS machine. The average values presented in the text were calculated from 3 independent liposomal preparations. We have added additional information on lines 169-172 to explain this.

For Figure 2 assessing effects of targeted and non-targeted liposome formulations, the authors need to add more n numbers. The data has high variability which casts a shadow on their interpretations.

We agree that there is some variability in individual placental characteristics observed in this preliminary study, as is common with any biological data set. We have changed the colours of the individual data points, so that each individual placenta is identifiable; this highlights the consistent lack of response to treatment between samples and we disagree that this normal level of variability casts any shadow on our interpretations. The levels of hCG, Ki67 and M30-positive cells we report are entirely consistent with studies from our group and others (refs 7, 18, 29). Furthermore, we have performed power calculations using these data which indicate that over 100 experiments would be required to detect differences between treatment and control groups. Finally, our previously published data (refs 7, 30) and the data sets in Figures 3, 4 and 5 consistently demonstrate a lack of detrimental effect of empty liposomes on human placental explants, supporting the findings in Figure 2. We have highlighted these points in the discussion (lines 484-488).

Did the authors assess what’s the rate/extent of uptake of the liposomal particles?

Although we did not directly assess the rate of uptake of the liposomes in this study, we have investigated this in first trimester and term human placental explants using immunofluorescence in our previous studies (7, 30)

In addition, maybe I missed this, but are the samples in figure 2 matched? i.e. sample placenta across the three treatment groups?

Yes; the samples in Figure 2 are matched. We have updated the figure and legend to reflect this, as described above.

N numbers for Figure 3, discussing effects on hCG and LDH secretion are missing.

We have now added n numbers to the legend of Figure 3.

Figure 3 shows two concentrations of free EGF only for FGR explants, but the authors do not really discuss the dosage or the reason for two dosages just for FGR in the text.

We have now added the following information in lines 203 - 207: “The EGF concentrations of 50 ng/ml and 100 ng/ml were chosen based on previously published data examining the effects of EGF on term placental explants (refs 18-20). A single concentration of 100 ng/ml EGF was used in healthy placental explants; whereas two concentrations of 50 ng/ml and 100 ng/ml were used in FGR explants to assess any possible concentration-dependent responses.”

To follow up on point 5 above, figure 4 C & D data sets only show one free FGR column. What concentration was this and why was it chosen?

We assume the reviewer means free EGF, not free FGR. As we saw no concentration dependent effect in Figure 3, we chose to use the highest concentration of EGF (i.e. 100ng/ml) in this experiment.  This has been clarified in the text (lines 370 – 372) and the axes and figure legends in Figures 3 and 4 have now been corrected.

In figure 5D, the authors suggest that FGR explants could be divided into responders and non-responders. What is the basis/logic for this conclusion? The controls in 5C also appear to split into two groups, so do the samples in Figure 4B, 4C and 4D. How do the authors explain this split in their samples? The high variability could explain the absence of significant changes in the samples due to EGF, but that cast a doubt upon the reliability of the targeted system.

We would like to clarify to the reviewer that in Figure 5C, almost all healthy explants have responded to treatment i.e. all data points lie above the 100% control line (shown in red). This is not the case for the FGR explants, where a significant number fall below the 100% control line; this is why we have concluded that the FGR placentas can be split into responders and non-responders (5D) but the healthy control explants (5C) all respond. We have now clarified this in the results (lines 411-413).

Regarding the data in Figure 4, we agree that these data show biological variability, but when the data are expressed as a % of matched control (i.e. how the data are presented in figure 5), the data do not separate into responders and non-responders. It is worth noting that FGR is a multifactorial disease, and placental dysfunction may present in many different ways, including changes in placental structure, cell turnover, hormone secretion, nutrient transfer capacity and/or vascular function. We do not believe that the variability within our data casts doubt upon reliability of the targeted delivery system, more that it reflects the ability of EGF to correct the underlying pathology in each individual placental explant. For example, if the FGR was the result of vascular defects and poor perfusion, rather than impaired cell turnover, application of EGF would likely not alter the basal rate of apoptosis. We have now expanded on the original text in the discussion (lines 502-509).

Lines 455-456; term cytotrophoblast spontaneously differentiate to syncytiotrophoblast in culture and do not need EGF stimulation.

We have reworded this sentence to state that “at term, the spontaneous differentiation of isolated cytotrophoblasts into syncytiotrophoblast can be enhanced by EGF (19, 35)” (lines 493-494).

The authors need a better and specific parameter for evaluating the effects EGF targeted delivery. The kinase array is not specific for EGF and assessing specific downstream activity might help support their conclusion.

We agree that the kinase array is not specific for EGF but it does identify a number of phosphorylated molecules downstream of EGFR and the specific signalling pathways that are activated (e.g. STATs, mTOR, PLC gamma-1). The reason we used this array was because we were unsure whether the targeted EGF liposomes would lead to classical activation of the EGF pathway, given the unconventional mode of delivery; we combined the phosphokinase array with bioinformatics analysis to better understand the downstream signalling pathways activated. However, we do certainly agree that further work is required to more comprehensively assess the downstream targets of our liposomal formulations. We have now clarified this in the text (lines 537-539; 552-555).

As the authors indicate the effects of EGF in explants has not been completely explored. Their results on ki67 staining- for proliferating cells and % apoptotic cells are at best confusing – for all three treatment modes. While the authors provide interesting data about specific delivery of the particles (and I am impressed with the immense work that went to it) their EGF targeting data fails to deliver the impact of their technology. Narrowing down to specific effects of EGF and evaluating those will significantly help improve the manuscript.

We agree  that more work is now needed to narrow down the specific effects of EGF (beyond the scope of the current study) and have now acknowledged this in the discussion (lines 553-554).

Reviewer 2 Report

The topic of this manuscript is very important, as the fetal growth restriction is a major health problem. Additionally, the placental-specific drug delivery is critical for patient safety. But, several concerns needs to be addressed to fit for publication as follows:

  1. Line 68-72: very long paragraph. Divide it for clarifying the meaning.
  2. Material and methods:
  • On what basis the authors have chosen the current EGF treatment regimen (the dose and route of administration). Justification with references is highly recommended.
  • What are the total number, initial weight, and age of mice used?
  • How the mice were anesthetized?
  • The legend of Table 1 is missed. The full term of all abbreviations used within the table should be clarified in the footnote.
  • The references of most methods are missed. E.g. immunohistochemistry.
  • Statistical analysis: what is meant by the data is presented as median? The data should be presented as means ± standard error or standard deviation. In addition, what is the statistical model used?
  1. Results:
  • The presentation of figures 2-5 is confusing in the present form. Thus, it is highly recommended to be presented as columns and the data presented as means ± standard error or standard deviation. Also, adding letters or figures denoting the significance compared to the control is highly recommended.
  • The data should be described as significant or non-significant.
  • Add the exact p-value of significance.
  • The full term of all abbreviations used should be clarified in the legend.
  • Several paragraphs should be transferred to the discussion E.g. lines 337-338, 368-370, 397-401.
  1. Discussion section is a major concern in the manuscript. In the present form, the discussion is a repetition of the introduction (lines 431-450) and results. It needs to be deepening by comparison to the earlier studies and more possible interpretations of the findings of the study.
  2. The use of abbreviations should be carefully revised throughout the manuscript. For instance, line 26, hCG should be mentioned as Human chorionic gonadotrophin (hCG) then the abbreviation should be used further. Also, line 54: replace fetal growth restriction with FGR. Such error has been repeated many times throughout the manuscript.

Author Response

Reviewer 2:

Line 68-72: very long paragraph. Divide it for clarifying the meaning.

This text has now been edited to form 2 sentences.

Material and methods:

On what basis the authors have chosen the current EGF treatment regimen (the dose and route of administration). Justification with references is highly recommended.

EGF was added to the culture medium of the human placental explants. The concentrations chosen (50 and 100 ng/ml) were based on previously published data examining the effects of EGF on term placental explants (refs 18-20). We have now added this justification and the supporting references to the text (lines 203 – 207).

What are the total number, initial weight, and age of mice used? How the mice were anesthetized?

Seven mice, aged between 8 - 11 weeks, with pre-pregnancy weights between 22.2 – 26.3g were used for the phage screening experiments shown in Figure 1. All other experiments were carried out using human placental tissue. Mice were anesthetised by an intraperitoneal injection of Avertin (150-250mg/kg). This information has now been added to the methods section (lines 93-94; 116-117).

The legend of Table 1 is missed. The full term of all abbreviations used within the table should be clarified in the footnote.

A legend and abbreviations have now been added to Table 1.

The references of most methods are missed. E.g. immunohistochemistry.

We have added additional references to the methods section.

Statistical analysis: what is meant by the data is presented as median? The data should be presented as means ± standard error or standard deviation. In addition, what is the statistical model used?

We disagree with the reviewer on this point. We have followed universally accepted statistical convention, which states that the appropriate statistical analyses are determined by whether the data to be analysed are parametric (normally distributed) or non-parametric (not normally distributed). Each of our datasets was subject to normality testing and the data were expressed and analysed accordingly. Parametric data (e.g. some of the patient demographic data) should be expressed as a mean +/- SEM and analysed with Student’s t tests and ANOVA. Non-parametric data (the majority of our data) should be expressed as medians and analysed by Mann Whitney U test for 2 groups or Kruskal Wallis for 3 or more groups). Please note that any normalised data (e.g. percentages, fold changes) are always non-parametric. 

Results:

The presentation of figures 2-5 is confusing in the present form. Thus, it is highly recommended to be presented as columns and the data presented as means ± standard error or standard deviation. Also, adding letters or figures denoting the significance compared to the control is highly recommended.

Our aim is to be as transparent as possible when presenting our data. As well as following statistical convention and presenting our non-parametric data as medians (as described above), we have chosen to present our data as dot plots, so that each individual data point can be visualised. This allows the normal biological variability within our data to be easily observed. If we were to present our data as columns in a bar graph, the variability would be concealed. We do not believe that this is good scientific practise, so we would prefer to present the data in its current format.

The data should be described as significant or non-significant.

We have now added this information into the relevant figure legends to highlight where observations are significant or non-significant.

Add the exact p-value of significance.

We have now added exact P values into the relevant figure legends to highlight statistical significance.

The full term of all abbreviations used should be clarified in the legends.

We have now updated the legends to include abbreviations.

Several paragraphs should be transferred to the discussion E.g. lines 337-338, 368-370, 397-401. Discussion section is a major concern in the manuscript. In the present form, the discussion is a repetition of the introduction (lines 431-450) and results. It needs to be deepening by comparison to the earlier studies and more possible interpretations of the findings of the study.

Given that the general readership of this journal, pharmaceutical scientists, will not be familiar with the unique intricacies of placental structure and function, we have included some extra background information within the results section. This is to justify experimental design and aid reader understanding and interpretation of our data; we believe it should remain. We have revised the discussion in places to add depth (lines 519-534); however, we lead the field in this specific area of research so there are few other directly relevant studies for us to compare our results to.

The use of abbreviations should be carefully revised throughout the manuscript. For instance, line 26, hCG should be mentioned as Human chorionic gonadotrophin (hCG) then the abbreviation should be used further. Also, line 54: replace fetal growth restriction with FGR. Such error has been repeated many times throughout the manuscript.

We have updated the text accordingly.

Round 2

Reviewer 1 Report

Accepted

Author Response

We would like to thank the reviewer once again for their excellent comments and suggestions.

Reviewer 2 Report

The manuscript is somewhat improved but the authors have not addressed some comments that have been previously mentioned as follows:
1.    The reference of the immunohistochemistry technique is still missed. As this technique is not the author's own work, the author should refer to the reference of this technique.
2.    The authors mentioned that "Given that the general readership of this journal, pharmaceutical scientists, will not be familiar with the unique intricacies of placental structure and function; we have included some extra background information within the results section. This is to justify experimental design and aid reader understanding and interpretation of our data". Scientifically and to avoid confusion to the readers, the extra background information should be clarified in the introduction section, not in the results section. In addition, the justification of the experimental design or any interpretations of the findings should be present in the discussion, not the results section.
